# Atom–Diatom Reactive Scattering Collisions in Protonated Rare Gas Systems

**DOI:** 10.3390/molecules26144206

**Published:** 2021-07-11

**Authors:** Debasish Koner, Lizandra Barrios, Tomás González-Lezana, Aditya N. Panda

**Affiliations:** 1Department of Chemistry, Indian Institute of Science Education and Research (IISER) Tirupati, Karakambadi Road, Tirupati 517507, Andhra Pradesh, India; debasishkoner@iisertirupati.ac.in; 2Department of Chemistry, CMS—Centre for Molecular Simulation, IQST—Institute for Quantum Science and Technology and Quantum Alberta, University of Calgary, 2500 University Drive N.W., Calgary, AB T2N 1N4, Canada; lizandra.barriosherr@ucalgary.ca; 3Instituto de Física Fundamental, IFF-CSIC, Serrano 123, 28006 Madrid, Spain; 4Department of Chemistry, Indian Institute of Technology Guwahati, Assam 781039, India; anp@iitg.ac.in

**Keywords:** reactive scattering, atom-diatom dynamics, quantum calculations, statistical dynamics

## Abstract

The study of the dynamics of atom–diatom reactions involving two rare gas (Rg) atoms and protons is of crucial importance given the astrophysical relevance of these processes. In a series of previous studies, we have been investigating a number of such Rg${}^{\left(1\right)}+$ Rg${}^{\left(2\right)}$H${}^{+}\to$ Rg${}^{\left(2\right)}+$ Rg${}^{\left(1\right)}$H${}^{+}$ reactions by means of different numerical approaches. These investigations comprised the construction of accurate potential energy surfaces by means of ab initio calculations. In this work, we review the state-of-art of the study of these protonated Rg systems making special emphasis on the most relevant features regarding the dynamical mechanisms which govern these reactive collisions. The aim of this work therefore is to provide an as complete as possible description of the existing information regarding these processes.

## 1. Introduction

The presence of rare gas (Rg) atoms and H${}^{+}$ in the early stages of the formation of the Universe and their abundance in the interstellar medium (ISM) [1,2,3,4,5,6,7] explains the interest on understanding the relevant features regarding the chemistry involving protonated Rg species. Thus, for example, the helium hydride ion, HeH${}^{+}$, is one of the first bond formed by radiative association in the primordial nucleosynthesis, playing a crucial role in the formation of molecular hydrogen. Its existence was predicted a long time ago, but it has been only in recent years that its detection has been finally confirmed [8]. The presence of ArH${}^{+}$ in Crab nebula has been inferred by spectra recorded by the Herschel Space Observatory [9]. Reactive collisions among these species are usually characterized by the absence of potential energy barriers and are therefore possible at the low temperature regime existing in the ISM. The study of the dynamics governing those processes can provide useful information about the evolution of the stellar medium as revealed by the growing interest of these systems in an astrophysical context [10,11,12].

Reactions of an Rg atom with hydrogen molecular ion, H${}_{2}^{+}$, are possibly among the most commonly studied processes of this kind due to their relevance in the early universe chemistry [3,13]. The dynamics of Rg + H${}_{2}^{+}\to$ RgH${}^{+}$ + H have been investigated in a series of works along the years [14,15,16,17,18,19,20,21,22,23,24]. Theoretical calculations are usually restricted to the ground electronic potential energy surface (PES) which exhibits a minimum along the reaction path. As a result, the reaction at low energy seems to proceed along collinear and near-collinear pathways mediated by the formation of an intermediate complex with numerous resonances observed in the corresponding probabilities [19,22,23]. Analogously, the possible formation of Rg${}_{2}$H${}^{+}$ species have been explored experimentally [25,26] and many theoretical investigations [27,28,29,30,31,32,33,34,35,36,37,38,39,40] suggest that the collinear [Rg–H–Rg]${}^{+}$ arrangement formed by means of the insertion of the proton into the weak bond between the Rg atoms is sufficient to make the complex stable [37,41,42]. A variety of different numerical approaches have been employed in the study of the dynamics of this type of processes. The development of PESs [14,17,21,43,44,45,46,47,48,49,50,51,52] for the different (RgH${}_{2}^{+}$) complexes was accompanied by a series of dynamical studies employing both quantum mechanical (QM) and quasi-classical trajectory (QCT) methods.

Early accurate ab initio calculation of structure and energies [40] for He${}_{2}$H${}^{+}$ were followed by the development of PESs [34,35] and the study of bound and quasibound states of the He${}_{2}$H${}^{+}$ and He${}_{2}$D${}^{+}$ complexes [34]. QM dynamical investigations of the He + HeH${}^{+}\to$ HeH${}^{+}$+He reaction by means of time-dependent wave packet (TDWP) calculations on these PESs [29,34,53] revealed noticeable differences between the probabilities obtained with the Coriolis coupled (CC) and centrifugal sudden (CS) approaches.

Besides the above-mentioned complexes formed solely with He and H${}^{+}$, examples of mixed Rg cations ${\left[{\mathrm{RgHRg}}^{\prime }\right]}^{+}$ and species with heavier Rg elements, from Ne to Xe, have been also studied in the past [26,38,41,42,54,55,56,57,58,59,60,61]. These investigations were completed with the case of lighter Rg atoms such as He, Ne or Ar [54,62,63,64,65].

In this work, we review theoretical state-of-the-art investigations on systems involving two Rg atoms and H${}^{+}$ performed by us with a number of methods. The studies involved the development of ab initio PESs describing the existing interactions between the colliding atoms and the application of different numerical techniques to study the dynamics of the process. Both QCT and QM approaches are comparatively applied within their different frameworks: Gaussian and histogram binning for the former and the already mentioned CC and CS alternatives for the latter. Given that these reactions are mediated by the presence of a relatively deep potential well in the intermediate region between reactants and products, a statistical quantum method (SQM) is also employed to test the possible importance of complex-forming mechanisms on the overall dynamics of the process. Our main goal is therefore to provide a complete compilation of recent studies on these reactions, thus giving a detailed description of how to tackle the dynamical analysis of this particular atom-diatom collisions.

The structure of this work is the following. In Section 2, we describe the procedure followed to obtain the PES, including first the ab initio methods (Section 2.1) and second the fitting to an analytical expression (Section 2.2). In Section 3, we discuss the details of the different methods employed to study the dynamics of these reactions. In Section 3.2, Section 3.3, Section 3.4, Section 3.5 and Section 3.6, we show the examples of the He + HeH${}^{+}\to$ HeH${}^{+}$ + He, He + NeH${}^{+}\to$ HeH${}^{+}$ + Ne, Ne + HeH${}^{+}\to$ NeH${}^{+}$ + He, Ne + NeH${}^{+}\to$ NeH${}^{+}$ + Ne and Ar + ArH${}^{+}\to$ ArH${}^{+}$ + Ar, respectively. Finally, in Section 4 concluding remarks are presented.

## 2. Potential Energy Surfaces

### 2.1. Methods

In the literature, protonated Rg systems have been investigated at different levels of theory. However, the “gold standard” coupled-cluster singles, doubles, and perturbative triples (CCSD(T)) method with augmented cc-pVnZ (n = T, Q, 5) basis sets provide the best results compared to other levels. For He${}_{2}$H${}^{+}$, [HeHNe]${}^{+}$, Ne${}_{2}$H${}^{+}$ and Ar${}_{2}$H${}^{+}$ systems, analytical PESs were constructed using ab initio energies obtained using CCSD(T) electronic structure calculations. While the d-aug-cc-pVTZ basis set was used for He${}_{2}$H${}^{+}$, the aug-cc-pVQZ basis set was chosen for all the other systems while carrying out the ab initio calculations. The ab initio calculations included the computation of energies for both the corresponding triatomic and three diatomic fragments.

For the construction of PESs for the triatomic system ABC, ab initio energies were computed along the grids defined in internal coordinates ($r_{\mathrm{AB}},r_{\mathrm{BC}}$ and θ), where $r_{\mathrm{AB}}$ and $r_{\mathrm{BC}}$ are the two internuclear distances between hydrogen and Rg atoms and θ is the angle between $r_{\mathrm{AB}}$ and $r_{\mathrm{BC}}$ as shown in Figure 1. Adiabatic PESs for ABC type systems can be represented as a many-body expansion function [66]: (1)$$\begin{array}{cc} V_{\mathrm{ABC}}\left(r_{\mathrm{AB}},r_{\mathrm{BC}},r_{\mathrm{AC}}\right) & =V_{A}^{\left(1\right)}+V_{B}^{\left(1\right)}+V_{C}^{\left(1\right)}+\\ \; & V_{\mathrm{AB}}^{\left(2\right)}\left(r_{\mathrm{AB}}\right)+V_{\mathrm{BC}}^{\left(2\right)}\left(r_{\mathrm{BC}}\right)+V_{\mathrm{AC}}^{\left(2\right)}\left(r_{\mathrm{AC}}\right)+V_{\mathrm{ABC}}^{\left(3\right)}\left(r_{\mathrm{AB}},r_{\mathrm{BC}},r_{\mathrm{AC}}\right). \end{array}$$

Here, $V_{\mathrm{ABC}}\left(r_{\mathrm{AB}},r_{\mathrm{BC}},r_{\mathrm{AC}}\right)$ is the potential energy of the total system, $V_{i}^{\left(1\right)}$s are the energies of the free atoms in their corresponding states, $V_{i}^{\left(2\right)}$s are two-body (2B) interaction energies, and $V_{\mathrm{ABC}}^{\left(3\right)}\left(r_{\mathrm{AB}},r_{\mathrm{BC}},r_{\mathrm{AC}}\right)$ is the three-body (3B) interaction energy.

To represent the 2B interaction energies, analytical PESs for all the possible diatomic fragments were obtained using either cubic spline interpolation or nonlinear curve fitting methods. For the nonlinear curve fitting, the diatomic terms were expressed using a polynomial form proposed by Aguado and Paniagua [67]:(2)$$V_{\mathrm{AB}}\left(r_{\mathrm{AB}}\right)=\frac{c_{0}e^{-\alpha r_{\mathrm{AB}}}}{r_{\mathrm{AB}}}+\sum_{i=1}^{M}c_{i}\rho_{\mathrm{AB}}^{i},$$
where *M* is the order of the polynomial, $\rho_{\mathrm{AB}}=r_{\mathrm{AB}}e^{-\beta_{\mathrm{AB}}r_{\mathrm{AB}}}$, $c_{i}$s are linear parameters, and α and β are nonlinear parameters.

The 3B interaction energy terms are expressed as [67]
(3)$$V_{\mathrm{ABC}}^{\left(3\right)}\left(r_{\mathrm{AB}},r_{\mathrm{BC}},r_{\mathrm{AC}}\right)=\sum_{i,j,k=0}^{M}d_{ijk}\rho_{\mathrm{AB}}^{i}\rho_{\mathrm{AB}}^{j}\rho_{\mathrm{AC}}^{k},$$
where $d_{ijk}$ are the linear parameters and $\rho_{i}=r_{i}e^{-\beta_{i}r_{i}}$. The constraints i+j+k≠i≠j≠k and i+j+k≤M are used in Equation (Equation 3) to make the 3B interaction energies zero at asymptotic limits. The Levenberg–Marquardt nonlinear optimization algorithm [68] was used to determine the linear and nonlinear parameters in Equation (Equation 3).

### 2.2. Analytical PESs

Global analytical PESs for RgH${}^{+}$, Rg${}_{2}$H${}^{+}$, or [RgHRg’]${}^{+}$ systems constructed from ab initio energies are reported in many instances in the literature. Details of electronic structure calculation methods and basis sets, number of ab initio energies used, and root mean square errors (RMSE) for the analytical PESs discussed in this review are tabulated in Table 1 as reported in the original references. A large number of ab initio energies are used in the fitting procedures except for He${}_{2}$H${}^{+}$. The fitting errors are less than 0.1 kcal/mol for all the cases, which guarantees high quality of those analytical PESs. For He${}_{2}$H${}^{+}$, apart from the PES reported in Table 1, another analytical PES has also been constructed using 15682 energies calculated at multi-reference configuration interaction and d-aug-cc-pV5Z level of theory [35] with an RMSE of 0.048 kcal/mol. Similarly, a global PES for Ar${}_{2}$H${}^{+}$ has been reported using 7040 QCISD/6-311++G(3df,3pd) energies with fitting RMSE of 0.143 kcal/mol [39].

Color maps of the analytical PESs reported in Table 1 are shown in Figure 2 for different possible reaction channels. Figure 2 shows the existence of deep potential well regions in all the systems for collinear and near collinear geometries with the hydrogen atom positioned between two Rg atoms. Geometries for the global minima for all the systems correspond to linear configurations, which are symmetric for Rg${}_{2}$H${}^{+}$ systems. Energies of the global minima and the possible reactant/product channels for the PESs are given in Table 2. The depth of the global minima from the nearest reactant asymptotes lie between ∼12–16.5 kcal/mol, with the lowest depth (12.014 kcal/mol) for [HeHNe]${}^{+}$ and highest depth for (16.605 kcal/mol) for Ne${}_{2}$H${}^{+}$. Hence, these triatomic cation systems are quite stable unlike noncovalent weak van der Waals complexes. The RgH${}^{+}$ bond distances for the equilibrium configurations of the triatomic systems and for the diatomic molecular ions are also given in Table 2. As it is observed, the internuclear separations between H and Rg atoms in the triatomic complexes get elongated compared to the free diatomic molecular ions. It is also seen that the He–H bond distance is longer in [HeHNe]${}^{+}$ than in He${}_{2}$H${}^{+}$. On the other hand, the Ne–H bond distance is longer in Ne${}_{2}$H${}^{+}$ compared to the bond in [HeHNe]${}^{+}$. This shows that size of the Ne atom plays an important role in this case.

In cases of Ne + HeH${}^{+}$ and Ar + ArH${}^{+}$ reactive systems, the existence of other interesting stable species is also observed in Figure 2. Shallow potentials can be seen in Ne+HeH${}^{+}$ and Ar+ArH${}^{+}$ color maps when the free Rg atom approaches the diatomic molecular ion from the other Rg atom side. This suggests formation of weak Rg⋯RgH${}^{+}$ complexes. The bond distance of one RgH is very close to the free RgH${}^{+}$. For [HeHNe]${}^{+}$, the second minimum (Ne⋯HeH${}^{+}$) is 1.36 kcal/mol lower than the Ne+HeH${}^{+}$ asymptote, whereas for Ar${}_{2}$H${}^{+}$, the other minimum (Ar⋯ArH${}^{+}$) is 2.74 kcal/mol lower in energy compared to the Ar+ArH${}^{+}$ asymptote. Although existence of a second minimum was reported for Ne${}_{2}$H${}^{+}$ in Reference [40], no such minimum was found for Ne${}_{2}$H${}^{+}$ in [70].

Minimum energy paths for possible atom+molecular ion collisions for these triatomic systems are shown for different internal angles in Figure 3. It can be seen that all the reactions are barrierless in their entrance channels. Collinear and near-collinear approaches are most favorable for these reactions and a potential well with depth ∼12–16.5 kcal/mol can be seen in the strong interaction region. However, for small attacking angles, barriers emerge in the same regions due to strong repulsion between the Rg atoms. The barrier height increases with the size of rare gas atoms, and with decreasing attacking angles.

## 3. Dynamical Studies

### 3.1. Methods

In this section, we summarize the most relevant aspects of the theoretical methods employed in the study of the title reactions.

#### 3.1.1. Time-Dependent Quantum Mechanical Method

Details of time-dependent (TD) wave packet (WP) methodology is well documented, and here we provide a brief outline only. In this method, the TD Schrödinger equation is solved on a grid using WPs. A set of body-fixed reactant Jacobi coordinates, as shown in Figure 1, is used in the calculations. An initial WP (either complex or real) prepared in the reactant asymptotic region is propagated using either the split operator [72] or Chebyshev real WP propagation [73,74] methods. The initial WP is constructed on equidistant grids along the radial (*R*,*r*) coordinates and Gauss–Legendre quadrature points along the angular degree of freedom. In this grid representation, the full Hamiltonian is written as a tridiagonal matrix as [75,76]
(4)$$\begin{array}{ccc} \hat{H} & = & \left[-\frac{\hbar^{2}}{2\mu_{R}}\frac{\partial^{2}}{\partial R^{2}}-\frac{\hbar^{2}}{2\mu_{r}}\frac{\partial^{2}}{\partial r^{2}}+\frac{j(j+1)}{2\mu_{r}r^{2}}+\frac{J\left(J+1\right)+j\left(j+1\right)-2K^{2}}{2\mu_{R}R^{2}}+V\left(R,r,\gamma \right)\right]\delta_{KK^{{}^{\prime }}}\\ & - & \frac{\hbar^{2}}{2\mu_{R}R^{2}}\lambda_{JK}^{+}\lambda_{jK}^{+}\sqrt{1+\delta_{K0}}\delta_{K+1,K^{\prime }}-\frac{\hbar^{2}}{2\mu_{R}R^{2}}\lambda_{JK}^{-}\lambda_{jK}^{-}\sqrt{1+\delta_{K1}}\delta_{K-1,K^{\prime }}.\; \end{array}$$

Here, V(R,r,γ) is the PES, *j* is the rotational quantum number of the reactant diatom, *J* is the total angular momentum of the reactants, and *K* is the projection of *J* on the body-fixed *z*-axis. The reduced masses along *R* and *r* coordinates are $\mu_{R}$ and $\mu_{r}$, respectively. λ in Equation (4) is defined as $\lambda_{JK}^{\pm }$ = $\sqrt{J(J+1)-K(K\pm 1)}$. Reactive scattering calculations are performed using either the full Hamiltonian as given in Equation (4) including the CC or within CS approximation where the off-diagonal terms in the Equation (4) are neglected, and only K=0 is considered.

In the WP propagation, the action of radial kinetic energy operators are evaluated using the fast Fourier transformation technique and the action of the angular momentum operator is evaluated in associated Legendre polynomial basis. After sufficient propagation, the energy dependent total reaction probabilities are calculated by summing the total flux passing through a fixed surface located at a large distance in the product channel. Spherical Hankel functions are used to compute the energy weightage of the initial translational WP.

#### 3.1.2. Time-Independent Quantum Mechanical Method

In order to study the reactive scattering for the proton exchange reactions within the time-independent approach, the ABC program [77] was used. The methodology for the time-independent quantum mechanical (TIQM) approach followed in the ABC package is documented in [77] and references therein. The time-independent Schrödinger equation is solved using the CC approach in hyperspherical coordinates. The diatomic ro-vibrational wavefunctions of all the available reactant/product channels for the given maximum energy and $j_{\max}$ (maximum number of rotational states allowed in each channel) are used to construct the CC basis functions. The CC hyper-radial equations are then solved by using a constant reference potential log derivative method between $\rho_{\max}$ to $\rho_{\min}$ (ρ is the hyper-radius) in $n_{\sec}$ sectors. Finally, the *S*-matrix elements for a particular *J* with diatomic parity (*p*) and triatomic parity (*P*) for all the channels are calculated from the final log derivative matrix by applying scattering boundary conditions.

#### 3.1.3. Quasiclassical Trajectory Calculation

The standard QCT methodology for atom-diatom collisions, discussed in details in [78,79,80,81,82], is followed to study the H${}^{+}$ transfer processes between Rg atoms. Initial conditions for the trajectories are sampled using standard Monte Carlo sampling in reactant Jacobi coordinates. The impact parameters in cross sections and rate constant calculations are sampled following either the normal procedures or using stratified sampling scheme [79,81,82]. Twelve coupled Hamilton’s equations of motion are then integrated numerically in reactant Jacobi coordinates using the fourth-order Runge–Kutta method. The total energy and total angular momentum were conserved up to sixth and tenth decimal places for all the trajectories. The ro-vibrational states for the reactant and product diatoms were determined following either the semiclassical theory of bound states or using the discrete variable representation based Colbert–Miller method [83]. The final quantum numbers of the product were assigned using both histogram binning (HB) and Gaussian binning (GB) methods [81,84,85].

The reaction probability for a selected initial ro-vibrational state and a given total angular momentum can be computed as
(5)$$P_{v,j,J}\left(E\right)=\frac{N_{r}}{N_{\mathrm{tot}}},$$
where $N_{r}$ is the number of reactive trajectories and $N_{\mathrm{tot}}$ is the total number of trajectories. The reaction cross section for a given initial ro-vibrational state is then calculated as
(6)$$\sigma_{v,j}\left(E,\theta \right)=\pi b_{\max}^{2}P_{v,j}\left(E\right)$$
where $b_{\max}$ is the maximum impact parameter for which reactive collision can occur.

The initial state selected differential cross sections (DCS) $\left(d\sigma_{r}/d\Omega \right)$ is computed as
(7)$$\frac{d\sigma_{r}\left(E\right)}{d\Omega }=\frac{1}{2k^{2}\sin\;\theta }\sum_{J=0}^{J_{\max}}\left(2J+1\right)\frac{N_{r}\left(J,\theta \right)}{N_{\mathrm{tot}}\left(J\right)}\;\frac{1}{\Delta \theta },$$
where $k=\sqrt{2\mu_{R}E/\hbar^{2}}$ and, $N_{r}\left(J,\theta \right)$ and $N_{\mathrm{tot}}\left(J\right)$ are the number of reactive trajectories scattered at an angle θ and the total number of trajectories run, respectively, for a given *J*. $N_{r}\left(J,\theta \right)$ can be determined by using histograms along θ.

#### 3.1.4. Statistical Quantum Mechanical Method

For those reactions which proceed via a complex-forming mechanism it is possible to apply statistical techniques. One of such methods is the statistical quantum mechanical (SQM) developed by Manolopoulos and coworkers [86,87]. A detailed description of the numerical details of this technique can be found in previous references [88,89,90,91,92,93,94], where the SQM approach has been employed to study atom-diatom reactions. Under the assumption of a complex-forming mechanism governing the overall dynamics of the process, the state-to-state reaction probability can be approximated as
(8)$$|S_{vjl,v^{\prime }j^{\prime }l^{\prime }}^{J}\left(E\right){|}^{2}≃\frac{p_{vjl}^{J}\left(E\right)p_{v^{\prime }j^{\prime }l^{\prime }}^{J}\left(E\right)}{\sum_{v^{\prime \prime }j^{\prime \prime }l^{\prime \prime }}p_{v^{\prime \prime }j^{\prime \prime }l^{\prime \prime }}^{J}\left(E\right)},$$
where *v*, *j*, and *l* refer to the diatomic vibrational, rotational, and orbital angular momentum quantum numbers, respectively, and *l* is the orbital angular momentum In the above expression, Equation (Equation 8), $p_{vjl}^{J}\left(E\right)$ is the capture probability or probability of forming the collision complex from the rovibrational state vjl at the reactant arrangement at the total angular momentum *J* and the energy *E*, while $p_{v^{\prime }j^{\prime }l^{\prime }}^{J}\left(E\right)$ is the probability for the collision complex to decay to the final $v^{\prime }j^{\prime }l^{\prime }$ state of the product channel respectively. Indexes in the denominator run for all energetically accessible states. The above expression for the reaction probability in Equation (Equation 8) reveals that the SQM approach does not provide any information regarding the amplitude of scattering matrix and therefore it can reproduce any possible existing resonance structure, yielding exclusively to an average value.

These capture probabilities are obtained by solving a set of CC equations for each arrangement with the form
(9)$$\Psi^{\prime \prime }=W\left(R\right)\Psi \left(R\right),$$
where the interaction matrix W(R) is expressed as
(10)$$W_{v^{\prime }j^{\prime }l^{\prime },vjl}^{J}\left(R\right)=\left[\frac{2\mu }{\hbar^{2}}\left(E_{vj}-E\right)+\frac{l(l+1)}{R^{2}}\right]\delta_{v^{\prime }v}\delta_{j^{\prime }j}\delta_{l^{\prime }l}+\frac{2\mu }{\hbar^{2}}V_{v^{\prime }j^{\prime }l^{\prime },vjl}^{J}\left(R\right),$$
being μ the 3B collision mass [95] and $E_{vj}$ is the rovibrational energy of the diatom (v,j) state. The interaction potential matrix $V_{v^{\prime }j^{\prime }l^{\prime },vjl}^{J}$ can be finally expressed in terms of the diatomic vibrational wavefunctions, vector-coupling coefficients and spherical harmonics as explained in Reference [86].

A computationally cheaper alternative to the CC scheme described in Equation (Equation 10) is the CS approximation where the coupled-channel equations are expressed in smaller sets for each value of *K*, the projection of the angular momentum on the atom-diatom axis, is obtained,
(11)$$W_{v^{\prime }j^{\prime },vj}^{JK}\left(R\right)=\left[\frac{2\mu }{\hbar^{2}}\left(E_{vj}-E\right)+\frac{l(l+1)}{R^{2}}\right]\delta_{v^{\prime }v}\delta_{j^{\prime }j}+\frac{2\mu }{\hbar^{2}}V_{v^{\prime }j^{\prime },vj}^{K}\left(R\right),$$
where
(12)$$l={\left[J\left(J+1\right)+j\left(j+1\right)-2K^{2}+1/4\right]}^{1/2}-1/2.$$

The solution of these equations is performed by means of a TIQM using the corresponding full ab initio PES within the region defined between asymptotic distances and a capture radius at which the collision complex is supposed to be formed. Therefore, the intermediate region where the PES for complex-forming processes usually display a relatively deep well is neglected. The above state-to-state probabilities are employed for the calculation of the corresponding integral cross sections (ICS) according to the following expression: (13)$$\sigma_{vj,v^{\prime }j^{\prime }}\left(E\right)=\frac{\pi }{gk_{vj}^{2}\left(2j+1\right)}\sum_{JKK^{\prime }}\left(2J+1\right)|S_{vjK,vjK^{\prime }}^{J}\left(E\right){|}^{2},$$
with $k_{vj}^{2}=2\mu \left(E-E_{vj}\right)/\hbar^{2}$, $E_{vj}$ being the energy of the initial rovibrational state vj of the reactant diatom and *g* the electronic degeneracy. Finally, the ICS of Equation (Equation 13) evaluated in the collision energy $E_{c}=E-E_{vj}$, is employed for the calculation of the rate constants as
(14)$$k_{vj,v^{\prime }j^{\prime }}\left(T\right)=\sqrt{\frac{8\beta^{3}}{\pi \mu }}\int_{0}^{\infty }\sigma_{vj,v^{\prime }j^{\prime }}\left(E_{c}\right)E_{c}\;e^{-\beta E_{c}}\;dE_{c},$$
where $\beta ={\left(k_{B}T\right)}^{-1}$.

The calculation of the differential cross section (DCS) by means of the SQM method requires an extra approximation due to the lack of information regarding the amplitude of the scattering matrix $S_{vj,v^{\prime }j^{\prime }}$ mentioned above in Equation (Equation 8):(15)$$\frac{d\sigma_{v,j;v^{\prime },j^{\prime }}}{d\Omega }\left(\theta ,E_{c}\right)≃\frac{1}{8k_{vj}^{2}}\frac{1}{\left(2j+1\right)}\sum_{JK^{\prime }K}{\left(2J+1\right)}^{2}{\left[d_{K^{\prime },K}^{J}\left(\pi -\theta \right)\right]}^{2}|S_{v,j,K;v^{\prime },j^{\prime },K^{\prime }}^{J}\left(E_{c}\right){|}^{2}.$$

As a consequence of this expression, the statistical angular distributions are therefore, symmetric, predicting equal peaks both at the forward and backward scattering directions.

### 3.2. He + HeH${}^{+}\to$ HeH${}^{+}$ + He

The He + HeH${}^{+}\left(v=j=0\right)\to$ HeH${}^{+}$ + He process was explored for the first time using the TDQM method [36] on the ab initio energy-based analytical PES developed in [34]. Reaction probabilities were computed for this reaction within the CS approximation which are shown in Figure 4 for some selected *J* values. It is worth mentioning that exact quantum dynamical simulations for this reaction has also been carried out by Xu and Zhang [29] using the PES constructed in [35]. To be consistent with the PESs mentioned in this review, exact quantum dynamics as well as QCT calculations have been performed, and total reaction probabilities obtained from TDQM-CC and QCT calculations are shown in Figure 4. The QM probabilities for J= 10 and 20 oscillate mostly around 0.2, but remain slightly larger at low energies. Few sharp peaks are obvious for *J* = 10 at low energies which suggests formation of a metastable He${}_{2}$H${}^{+}$ complex in the potential well (see Figure 2 and Figure 3). Probability curves for the high *J*s are relatively smoother with broad peaks. It was observed in [29] that the TDQM CS probabilities for this reaction computed using References [34,35] PESs are very similar which is also obvious in this work for the TDQM-CC probabilities when compared to the results presented in [29]. This suggests that although different level of theories were used to calculate the two PESs, their global topology does not differ significantly. As can be seen in Figure 4, the QCT method nicely reproduces the overall behavior of the exact QM results. Results obtained from both QCT binning schemes are quite similar. However, due to inherent zero point energy leakage in QCT, finite probabilities for this reaction are predicted for energies below threshold for higher *J*s.

Total ICSs as a function of collision energy and rate constants as a function of temperature for the He + HeH${}^{+}\left(v=j=0\right)\to$ HeH${}^{+}$ + He reaction are shown in Figure 5. TDQM-CS results in this figure are taken from Reference [36]. The ICSs for this reaction have high values at low collision energies, but their magnitudes decrease with the increase of collision energies. The TDQM-CS method underestimates the ICSs in the entire energy range while results seem to be overestimated by both QCT approaches up to energies about 0.3 eV. Beyond that energy, the two QCT alternatives agree quite well in reproducing the ICSs. QM ICSs, specially those obtained with the TDQM-CC approach, exhibit many oscillations as a function of the energy. Rate constants obtained using different dynamical methods for He + HeH${}^{+}\left(v=j=0\right)\to$ HeH${}^{+}$ + He reaction are found to be independent of temperature beyond 250 K, following the classical Langevin capture model [96,97] for a barrierless ion-molecule reaction. While the TDQM-CS method underestimates the reaction rate constants, predictions obtained with the QCT approaches remain above the QM results. It is also observed that both QCT binning schemes produce very similar results except at low temperatures.

### 3.3. He + NeH${}^{+}\to$ HeH${}^{+}$ + Ne

One the characteristic features of reactions mediated by the presence of relatively deep potential wells is the existence of numerous resonance peaks in the probabilities as a function of the energy. In this particular case, the PES for the He + NeH${}^{+}$ reaction exhibits a well of ~16.14 kcal mol${}^{-1}$ for the ${\left[\mathrm{HeHNe}\right]}^{+}$ species between reactants and products [53,69]. As shown in Figure 6 for the case of the He + NeH${}^{+}\to$ HeH${}^{+}$ + Ne reaction, even for relatively large values of the total angular momentum, such as J=30 and 50, the WP results from in [53] exhibit noticeable maxima attributed to the formation of an intermediate complex supporting a number of quasi bound and resonance states. The cases shown in Figure 6 correspond to two different initial reactant states, in particular, NeH${}^{+}\left(v=0,j=0\right)$ and NeH${}^{+}\left(v=0,j=1\right)$. Peaks, which are still narrow at J=30, become however progressively broader as the value of the angular momentum increases, as revealed from the comparison with the case of J=50.

Differences between the CC and CS schemes for the WP calculation are seen for J=30, but it is clear from Figure 6 that this effect is more important as *J* gets larger. The statistical predictions obtained by means of the SQM approach, also included in the figure for comparison, reveals certain independence with respect to the specific framework considered, as the CS and CC results do not seem too different. The description provided by the SQM values of the WP reaction probabilities remain as an acceptable average for J=30 with no information regarding the resonance peaks, but for J=50 the CC (CS) prediction overestimate (underestimate) the corresponding WP results.

The threshold for reaction is ≈0.29 eV for the case of the ground rovibrational state NeH${}^{+}\left(v=0,j=0\right)$, a value which is increased for high values of the total angular momentum due to the increase of the centrifugal barriers. For this vibrationless case, thresholds exhibited by the QM-CC and QM-CS probabilities are different, and Figure 6 shows how the CS results have larger threshold energies. This feature is however not observed in the statistical case, with almost identical reaction probabilities for the SQM-CC and SQM-CS cases.

A similar comparison between QM WP and statistical predictions estimated by means of the SQM approach can be established through the opacity functions, that is, the reaction probability as a function of the total angular momentum *J* at a specific value of the collision energy. More precisely, Figure 7 shows the partial wave contribution $\left(2J+1\right)P^{J}\left(E_{c}\right)$ as a function of *J* for the initial-state selected cases (v=0,j=0) and (v=1,j=0) at $E_{c}=350$ meV and 450 meV. As seen in Figure 6 for the case of the energy, the WP probabilities exhibit also an oscillatory behaviour with respect to *J*. Whereas significant differences are observed between WP-CC and WP-CS results, the SQM predictions show little dependence on the specific scheme, either CC or CS, employed in the calculations. The statistical estimations for the reaction initiated from the ground rovibrational state NeH${}^{+}\left(v\;=\;0,j\;=\;0\right)$ provide a good average description of the corresponding QM WP probabilities. Furthermore, good agreement is also observed for the case of v=1, with the SQM probabilities providing a correct reproduction of the existing decrease of the opacity functions at J∼60 displayed by the QM-CS results. Partial waves coming from the larger *J* seem to contribute more noticeably in the QM-CC calculation rather than in the QM-CS approach where contributions coming from lower values of the total angular momentum are significant.

The present theoretical analysis of the dynamics of the He + NeH${}^{+}\left(v,j\right)\to$ Ne + HeH${}^{+}$ reaction includes the calculation of the ICSs for the (v=0,j=0) and (v=1,j=0) cases. In Figure 8, cross sections obtained with the WP-CS and WP-CC approaches are compared with the corresponding statistical predictions for a collision energy up to 0.5 eV. As suggested by the different comparison seen for the reaction probabilities (see Figure 7) for both initial rovibrational states between the WP and the SQM methods, the agreement for the (v=0,j=0) case contrasts with the differences observed with (v=1,j=0). Thus, whereas the energy threshold for the cross section when the reaction proceeds from the ground rovibrational state is well reproduced, the behaviour at the low energy regime exhibiting the characteristic trend of a barrierless process in the (v=1,j=0) case is clearly underestimated by the statistical prediction.

### 3.4. Ne + HeH${}^{+}\to$ NeH${}^{+}$ + He

The existence of a relatively deep well (∼18.45 kcal mol${}^{-1}$) in the PES for this reaction also leads to the presence of narrow resonances in the corresponding probabilities. Examples of such reaction probabilities for three different values of *J* and three possible initial states HeH${}^{+}\left(v,j\right)$: (v=0,j=0), (v=0,j=1), and (v=1,j=0) obtained with a WP method [63] are shown in Figure 9. Results from a QCT calculation, also included in the figure, reproduce in all cases the average trend of the WP probabilities although no information regarding the resonance peaks is recovered. The SQM predictions, in turn, despite to provide the correct threshold for reaction, overestimate noticeably the WP results as the energy increases for J=10 and 40. For the highest value of the total angular momentum shown in Figure 9, the SQM probabilities seem to remain below both the WP and QCT results. These discrepancies between QM results and the corresponding statistical predictions are usually interpreted as deviations of the overall dynamics of the reaction from a purely complex-forming process [90,91,98,99].

The oscillatory behaviour of the time-dependent QM (TDQM) probabilities of Figure 9 disappears when we obtain the ICSs, washed out as a result of the partial waves averaging effect. The corresponding cross section for the (v=0,j=0) case between 10${}^{-3}$ eV and 0.5 eV is shown in Figure 10. The comparison with the QCT and SQM results, also shown in the figure, reveals significant discrepancies with these two approaches at the lower energies [63]. The trend followed by the TDQM ICSs as the energy decreases is a consequence of the difficulties of the WP techniques at this regime, but as shown in [64], it can be solved by means of a TIQM calculation. Computationally much cheaper, it is possible to show by means of the SQM approach that the proper trend exhibited by the ICSs at such low-energy regime is directly related with a correct description of the asymptotic region employing sufficiently large distances in the calculation [63,88,100,101]. The ICS obtained with the TIQM method is in a nice agreement with both the QCT and SQM predictions.

The rate constants between T=10 K and 1000 K obtained from the ICSs shown in Figure 9 are presented in Figure 11 for the three initial rovibrational states of the reactant HeH${}^{+}$ under consideration. The comparison of the WP values reveals that the ground state (v=0,j=0) yields the larger k(T), followed by the one for the first rotationally excited state (v=0,j=1). The lowest rate constant in the comparison shown in Figure 11 is obtained for the first vibrationally excited states (v=1,j=0). In particular, at T=100 K, $k_{v=0,j=0}=6.67\times 10^{-10}$ cm${}^{3}$ s${}^{-1}$, whereas $k_{v=0,j=1}=5.81\times 10^{-10}$ cm${}^{3}$ s${}^{-1}$ and $k_{v=1,j=0}=4.49\times 10^{-10}$ cm${}^{3}$ s${}^{-1}$. The same sequence is observed among the corresponding SQM rates, but, the QCT calculation, on the contrary predicts certain crossing between the $k_{v=0,j=1}$ and $k_{v=1,j=0}$ rates around T∼130 K.

State-to-state rovibrational distributions of the Ne + HeH${}^{+}\left(v,j\right)\to$ He + NeH${}^{+}\left(v^{\prime },j^{\prime }\right)$ reaction for specific values of the collision energy have been also obtained by means of the theoretical approaches discussed in this review. In particular, final state resolved cross sections at two energies, 100 meV and 500 meV, for the reaction initiated from the ground rovibrational HeH${}^{+}\left(v=0,j=0\right)$ state obtained by means of TIQM, SQM, and QCT calculations are compared in Figure 12. Significant discrepancies with the QM result for the production of NeH${}^{+}$ in its vibrationaless state $v^{\prime }=0$ are observed at $E_{c}\;=\;100$ meV: On the one hand, the statistical prediction displays a maximum peak for the central rotational states NeH${}^{+}\left(v^{\prime }=0,j^{\prime }∼6\right)$, and on the other hand, the QCT distributions peaks at too low final rotational state $j^{\prime }=2$ in comparison with the TIQM values.

As the energy increases, on the contrary (see the case of $E_{c}=500$ meV at the bottom panel of Figure 12), both SQM and QCT calculations provide a fairly good counterpart of the TIQM rotational distributions for all possible product vibrational states NeH${}^{+}\left(v^{\prime }=0-2,j^{\prime }\right)$. The oscillations of the TIQM and QCT rotational distributions as a function of the rotational states for the production of vibrationless NeH${}^{+}\left(v^{\prime }=0\right)$ contrast however with the smooth trend followed by the statistical predictions.

The comparison of the different theoretical calculations for the DCS also reveals some differences with the energy variations. In Figure 13, results of TIQM, SQM, and two alternatives (GB an HB) for QCT calculations are compared for the same two collision energies considered for the rotational distributions (see Figure 12). The TIQM DCSs exhibit some asymmetry between the forward (θ∼0 degrees) and backward (θ∼180 degrees) scattering directions, a feature which is only described by the QCT calculation at 500 meV (see bottom panel of Figure 13). The imposed-by-construction forward-backward symmetry of the statistical approach fails therefore to give a proper description of the QM result, although both the forward peak at $E_{c}=100$ meV and the backward peak at $E_{c}=500$ meV.

The two options within the QCT calculation, the GB and HB, lead to similar DCS at both collision energies. The comparison with the TIQM result reveals a nice agreement with the TIQM distribution for the higher energy (500 mev) and noticeable discrepancies for the lower energy ($E_{c}=100$ meV).

### 3.5. Ne + NeH${}^{+}\to$ NeH${}^{+}$ + Ne

Also mediated by the presence of a potential well in the intermediate region, ${\left[\mathrm{NeHNe}\right]}^{+}$, of ~16.60 kcal mol${}^{-1}$ between reactants and products, the probabilities for the Ne + NeH${}^{+}\to$ NeH${}^{+}$ + Ne reaction display a rich structure of narrow resonances, especially at the low energy regime (see to panels of Figure 14). The SQM predictions remain on a constant value of 0.5, as a consequence of having equal channels for both reactants and products. Despite thresholds for reactions seem to be well described with this approach, the statistical value constitutes a poor average description of the WP reaction probabilities, with the only exception, perhaps of the largest value of the total angular momentum considered in the figure, J=120.

The comparison with the QM calculations becomes much more favourable for the statistical predictions in the case of the ICS. Figure 15 shows the cross sections for the SQM, TIQM and TDWP approaches in the collision energy between 1 meV and 0.4 eV when the reaction is initiated from the ground rovibrational state NeH${}^{+}\left(v=0,j=0\right)$ The already mentioned difficulties of the WP methods at relatively low energies here manifested in an spurious behaviour of the reaction probability as the energy decreases sufficiently (below ∼0.02 eV). The statistical cross sections, despite an overall qualitatively fair agreement with the TIQM result, remain below up to $E_{c}=0.2$ eV.

These quantitative discrepancies seen in the ICSs have also a consequence in the corresponding rate constants. In Figure 16, values of k(T) obtained with the TIQM and SQM cross sections of Equation (15) are compared between 50 K and 900 K. The TIQM rate constants are always larger than the statistical counterparts and, for example, at T=900 K, $k_{\mathrm{TIQM}}\left(T\right)=4.46\times 10^{-10}$ cm${}^{-3}$ s${}^{-1}$, whereas the SQM yields a value of $3.82\times 10^{-10}$ cm${}^{-3}$ s${}^{-1}$. The constant Langevin value of $4.63\times 10^{-10}$ cm${}^{-3}$ s${}^{-1}$ , also included in Figure 16 for comparison, remain above.

### 3.6. Ar + ArH${}^{+}\to$ ArH${}^{+}$ + Ar

This reaction is investigated using quantum and classical dynamical simulations on the analytical PES in [71]. Reaction probabilities for Ar + ArH${}^{+}\left(v=0,j=0\right)$→ ArH${}^{+}$ + Ar obtained from TDQM, TIQM and QCT methods are shown in Figure 17 for few selected *J*s. Numerous resonances are noticeable in the QM probability curves at low collision energies for J=0. As can be seen in Figure 2 and Figure 3, the potential for Ar${}_{2}$H${}^{+}$ contains a 15.55 kcal/mol deep well for linear and near collinear Ar–H–Ar geometries which supports the numerous meta-stable collision complexes during the proton exchange reaction. It has been reported in [71] that these intermediate species have average lifetime of ∼0.7 ps determined from the width of the resonances. The probability curves are comparatively smoother for high *J* values which suggests stripping mechanisms for the reaction.

As can be seen in Figure 17, the agreement between the TIQM and TDQM results is excellent except for the lowest collision energies. These differences originate from the finite time propagation of the WP in the TDQM method. Reaction probabilities calculated using the QCT simulations are also shown in Figure 17 and the results are in a fair agreement with the QM results. For *J* = 0 and 100, QCT-HB probabilities lie between 0.4 and 0.6, and the curves are rather flat while the QCT-GB probabilities are larger than QCT-HB probabilities in the low-energy region. However, both GB and HB schemes produce similar results for collision energies higher than 0.35 eV for low values of *J*s. The QCT method largely underestimates the QM probabilities at collision energies ∼0.15–0.4 eV, whereas those are in good agreement beyond 0.4 eV except for J=200. For *J* = 300, the QCT-HB method successfully reproduces the overall behavior of the QM probabilities while QCT-GB method slightly overestimates the QM results.

The effect of vibrational excitation on the Ar + ArH${}^{+}\to$ ArH${}^{+}$ + Ar reaction was also investigated in [71] using TDQM method. It was found that the vibrational excitation of ArH${}^{+}$ greatly reduces the reactivity for this reaction which is similar to the results for Ne + HeH${}^{+}$ reaction.

Total ICSs and rate constants for the Ar + ArH${}^{+}\left(v=0,j=0\right)$→ ArH${}^{+}$ + Ar reaction computed using TIQM and QCT methods in Reference [71] are given in the left panel of Figure 18. The cross sections are very large at low collision energies and decrease rapidly as $E_{c}$ increases reaching a plateau region. The overall behavior of the cross sections is quite similar to the results obtained for Ne + NeH${}^{+}$. Resonances present in the probability curves are mostly washed out due to the *J* averaging, and only a few broad peaks can be seen at very low energies. QCT-GB scheme reproduces the QM ICSs quite accurately up to 0.1 eV of collision energy but underestimates the QM results for E> 0.1 eV. The QCT-HB cross sections have a constant slope in the log-log plot which suggests a simple capture type cross sections typical for ion-molecule reactions.

Initial state selected rate constants for the proton exchange reaction between two Ar atoms with the reactant ArH${}^{+}$ in its ground ro-vibrational state are shown in the right panel of Figure 18. The QM rate constant increases with increase in temperature and becomes almost constant with values $∼6\times 10^{-10}$ cm${}^{3}$s${}^{-1}$ at T>600 K. The QCT-GB results follow a similar trend whereas QCT-HB rate constants remain almost constant at $∼6\times 10^{-10}$ cm${}^{3}$s${}^{-1}$ in the entire temperature range. The rate constant calculated following the Langevin’s capture model [96,97] for ion–molecule reaction is $∼6.88\times 10^{-10}$ cm${}^{3}$s${}^{-1}$ which is slightly higher than those obtained from dynamical simulations. However, it was found from QCT trajectories that only a fraction of trajectories which form collision complexes leads to products via proton exchange reaction and others produce reactants again.

## 4. Conclusions

Reactions in protonated rare gas (Rg) systems have been intensely investigated in recent years due to their astrophysical interest. In this review, we present results of previous studies on the reactions He + HeH${}^{+}$→ HeH${}^{+}$ + He; Ne + NeH${}^{+}$→ NeH${}^{+}$ + Ne; Ar + ArH${}^{+}$→ ArH${}^{+}$ + Ar; He + NeH${}^{+}$→ HeH${}^{+}$ + Ne and Ne + HeH${}^{+}$→ NeH${}^{+}$ + He. Different observables such as reaction probabilities, cross sections, and rate constants for those reactions have been obtained by means of exact quantum and quasi-classical trajectory (QCT) calculations on high-level ab initio energy-based analytical potential energy surfaces (PESs). The existence of a relatively deep well in the intermediate region between reactants and products enable as well the use of a statistical quantum method (SQM) which provides a fairly good average description of the overall dynamics. Different approximations introduced to both the time-dependent and time-independent quantum and statistical techniques such as the centrifugal sudden (CS) option have been successfully tested and alternatives for the binning procedure in the QCT such as the histogram and Gaussian approaches were comparatively employed. In general, all methods are found to be adequate for the study of the title processes although the CS approximation can introduce significant discrepancies with respect to the more rigorous coupled channel scheme. Moreover, the description provided by the time dependent techniques can be affected by some deficiencies of the wave packet propagation at the low energy regime. We think that one of the challenges in this area is the comparative analysis of these reactions when heavier Rg atoms are considered. It would be interesting, for instance, to see whether or not in these cases, the complex-forming mechanisms play a major role on the overall dynamics of the process, thus justifying the use of statistical techniques such as the SQM discussed here. Finally, we conclude that this complete sort of investigations which include as a fundamental first step the development of full ab initio PES represent a convenient strategy to understand the dynamics of the title reactions.

## Figures and Tables

**Figure 1 molecules-26-04206-f001:**
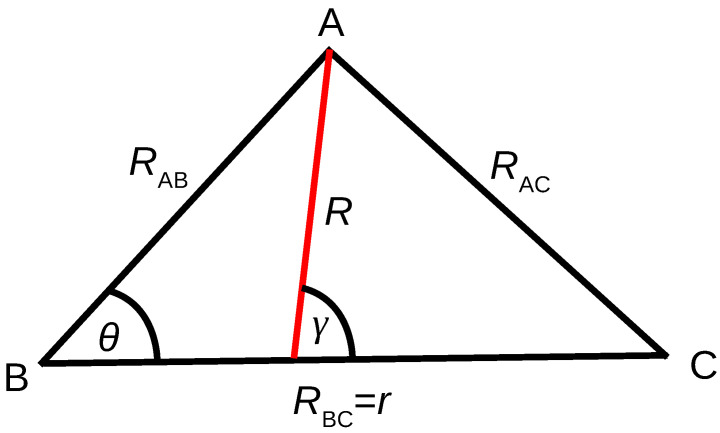
Coordinate considered for ab initio electronic structure calculations and dynamical simulations. $r_{\mathrm{AB}}$, $r_{\mathrm{BC}}$, and $r_{\mathrm{AC}}$ are the distances between atoms A and B, B and C, and A and C, respectively. θ is the angle between $r_{\mathrm{AB}}$ and $r_{\mathrm{BC}}$. While the set $r_{\mathrm{AB}}$, $r_{\mathrm{BC}}$ and θ is used for the ab initio points, the Jacobi coordinates set of *R*, *r* and γ is used for the propagation, where *R* is the distance from the center-of-mass of BC to A and γ is the angle between *R* and *r*.

**Figure 2 molecules-26-04206-f002:**
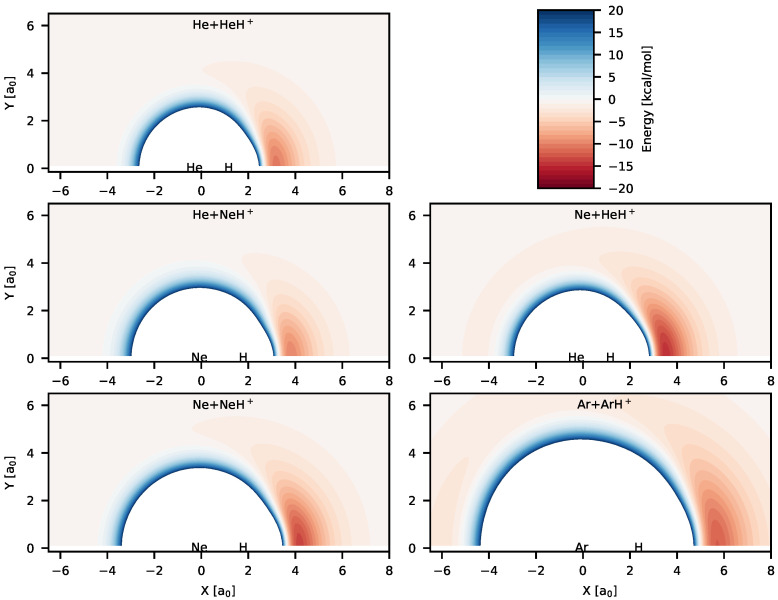
Color maps of the potential energies computed from the analytical PESs for different reactive systems. The diatoms are at their equilibrium geometry and placed on the “X” axis and the origin of each plot is set to the center-of-mass of the diatoms. The zero of energy is set to the asymptotic energy of the reactant channels for each plot. The blue regions have positive energies while the red regions have negative energies. Unique color-box is used for all the panels and is shown in top-right panel.

**Figure 3 molecules-26-04206-f003:**
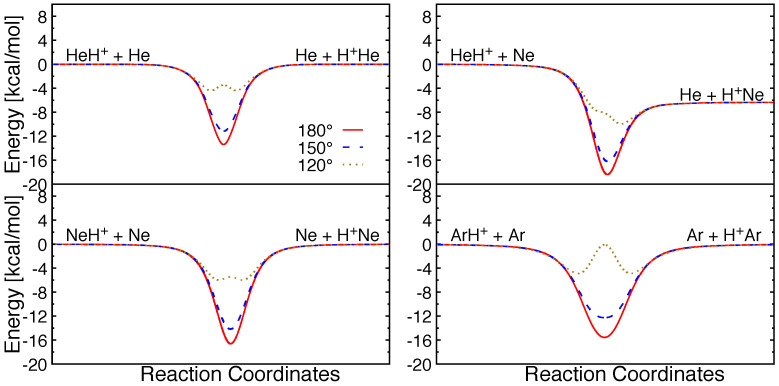
Minimum energy pathways along different attacking angle (Rg-H-Rg${}^{\prime }$) for different reactive collision systems.

**Figure 4 molecules-26-04206-f004:**
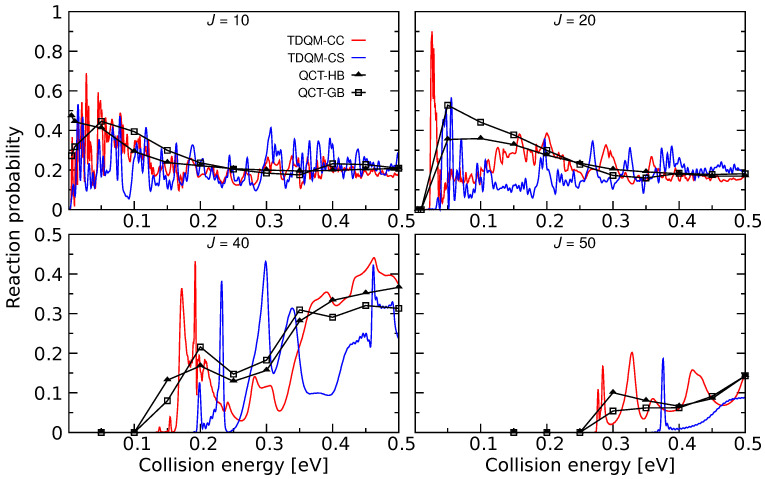
Total reaction probabilities as a function of collision energies for the He + HeH${}^{+}\left(v=0,j=0\right)$→ HeH${}^{+}$ + He for J=10, 20, 40 and 50 computed by means TDQM-CC (red solid lines), TDQM-CS (blue solid lines), and QCT (QCT-HB: black lines + filled triangles, QCT-GB: black lines+open squares) methods. TDQM-CS data are taken from Ref. [36].

**Figure 5 molecules-26-04206-f005:**
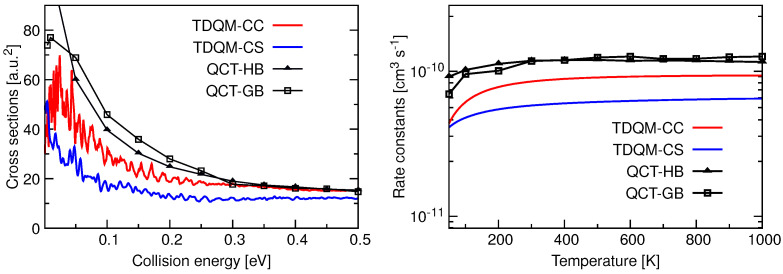
Total ICSs as a function of collision energy (**left panel**) and rate constants as a function of temperature (**right panel**) for the He + HeH${}^{+}\left(v=0,j=0\right)$→ HeH${}^{+}$ + He calculated following TDQM CC (solid red solid line) and TDQM CS (blue solid line) and QCT (QCT-HB: black lines+filled triangles, QCT-GB: black lines+open squares) methods.

**Figure 6 molecules-26-04206-f006:**
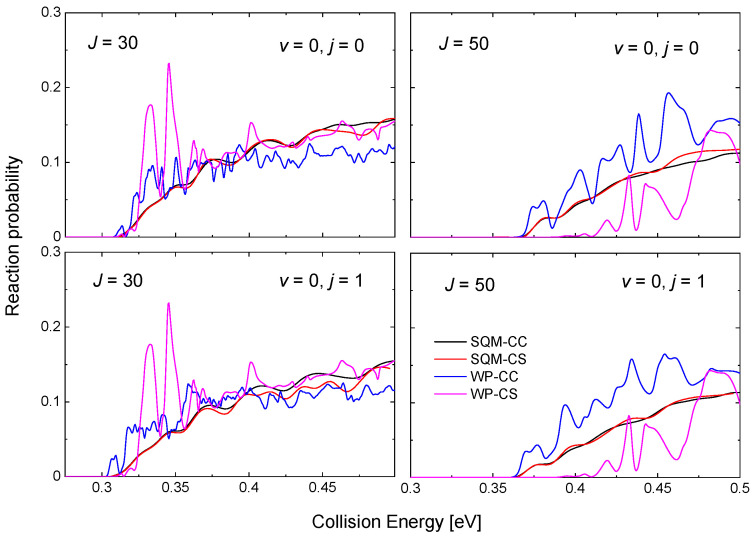
Reaction probabilities as a function of the collision energy (in eV) for the He + NeH${}^{+}\left(v,j\right)$ reaction for v=0,j=0 (**top panels**) and v=0,j=1 (**bottom panels**) and J=30 (**left panels**) and J=50 (**right panels**). WP results within the coupled-channel (blue line) and close-coupling schemes (magenta line) in comparison with the results from the SQM calculation within the CC (black line) and the CS (red line) schemes. Results adapted from the work in [53].

**Figure 7 molecules-26-04206-f007:**
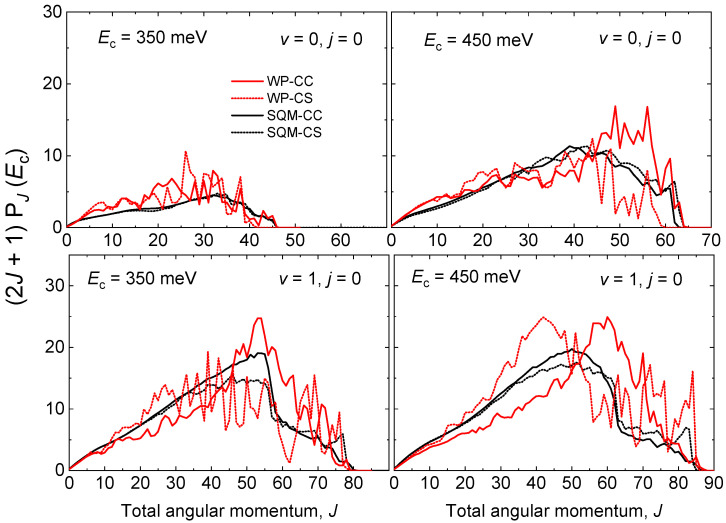
Opacity functions as (2J+1)P(J) for the He + NeH${}^{+}\left(v,j\right)$ reaction for v=0,j=0 (**top panels**) and v=1,j=0 (**bottom panels**) at two different values of the collision energy $E_{c}=350$ meV (**left panels**) and $E_{c}=450$ meV (**right panels**) as a function of the total angular momentum *J*. WP results (red lines) and SQM (black line) within the coupled-channel (solid line) and close-coupling schemes (dotted line) adapted from Reference [53] are compared.

**Figure 8 molecules-26-04206-f008:**
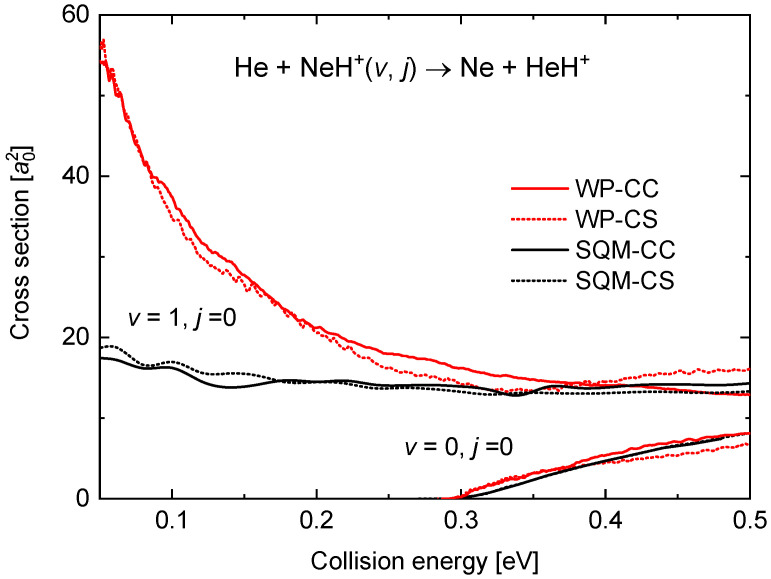
Integral cross sections (in bohr${}^{2}$) as a function of the collision energy in eV for the He + NeH${}^{+}\left(v,j\right)$ reaction. Results for (v=0,j=0) (in the bottom right corner) are compared with those for (v=1,j=0). WP results (red lines) and SQM (black line) within the coupled-channel (solid line) and close-coupling schemes (dotted line) are from the work in [53].

**Figure 9 molecules-26-04206-f009:**
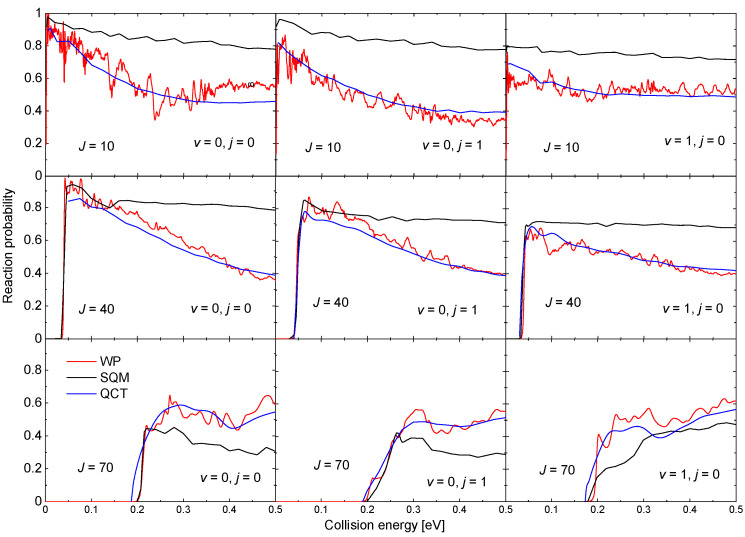
Reaction probabilities as a function of the collision energy (in eV) for the Ne + HeH${}^{+}\left(v,j\right)\to$ He + NeH${}^{+}$ reaction for v=0,j=0 (**left panels**), v=0,j=1 (**middle panels**) and v=1,j=0 (**right panels**) for different values of the total angular momentum: J=10 (**top panels**); J=40 (**second row panels**) and J=70 (**bottom panel**). Comparison of WP (red lines), SQM (black lines), and QCT (blue lines) results adapted from the work in [63].

**Figure 10 molecules-26-04206-f010:**
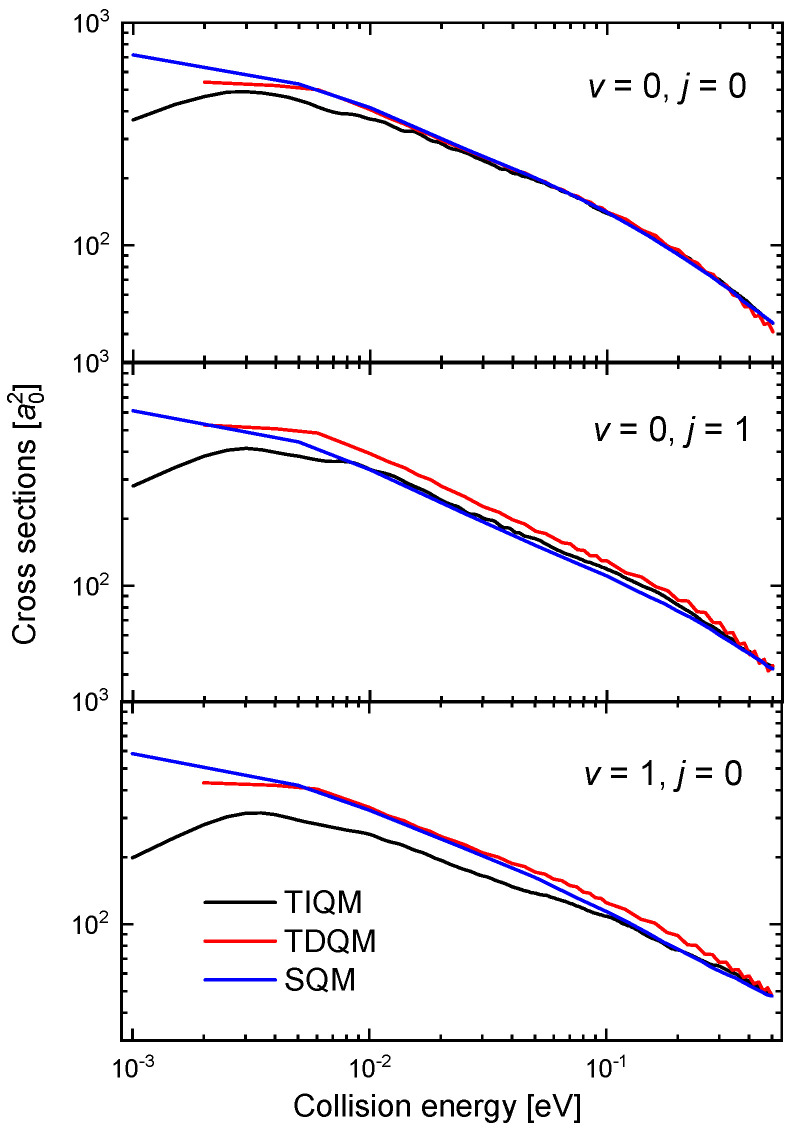
ICSs (in $a_{0}^{2}$) as a function of the collision energy (in eV) for the Ne + HeH${}^{+}\left(v=0,j=0\right)\to$ He + NeH${}^{+}$ reaction with a TDQM (red lines), TIQM (black line), SQM (blue lines) and QCT (open circles) calculations from Refs. [63,64].

**Figure 11 molecules-26-04206-f011:**
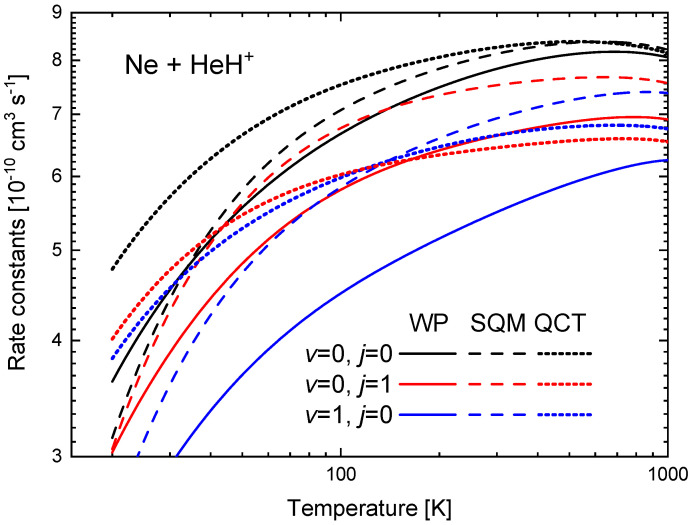
Rate constants (in $10^{-1}$ cm${}^{3}$ s${}^{-1}$) as a function of the temperature (in K) for the Ne + HeH${}^{+}\left(v,j\right)\to$ He + NeH${}^{+}$ reaction for v=0,j=0 (black), v=0,j=1 (red) and v=1,j=0 (blue) obtained with the WP (solid lines), SQM (dashed lines) and QCT (dotted lines) calculations of Reference [63]. Logarithmic scales have been used in both axes.

**Figure 12 molecules-26-04206-f012:**
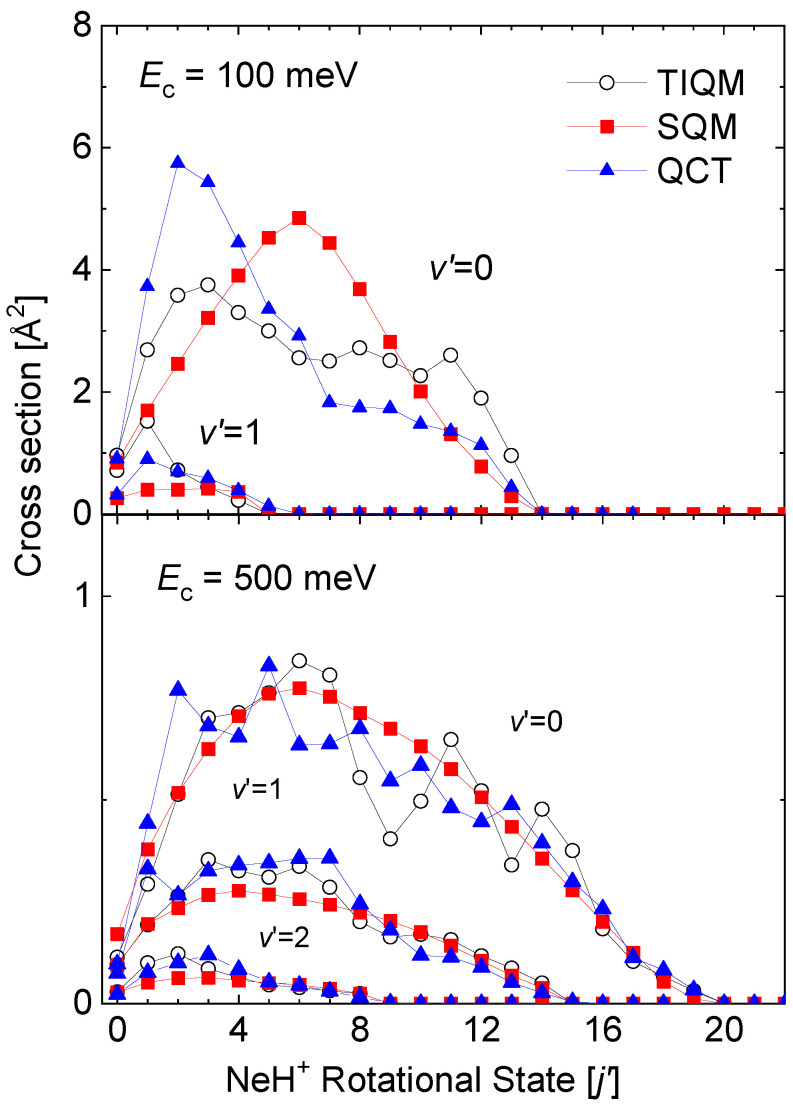
Rotational distributions (in Å${}^{2}$) for the Ne + HeH${}^{+}\left(v=0,j=0\right)\to$ He + NeH${}^{+}\left(v^{\prime },j^{\prime }\right)$ reaction at two different values of the collision energy: 100 meV (**top panel**) and 500 meV (**bottom panel**). TIQM (open black circles), SQM (red squares), and QCT (full blue triangles) cross sections from Ref. [64] are compared.

**Figure 13 molecules-26-04206-f013:**
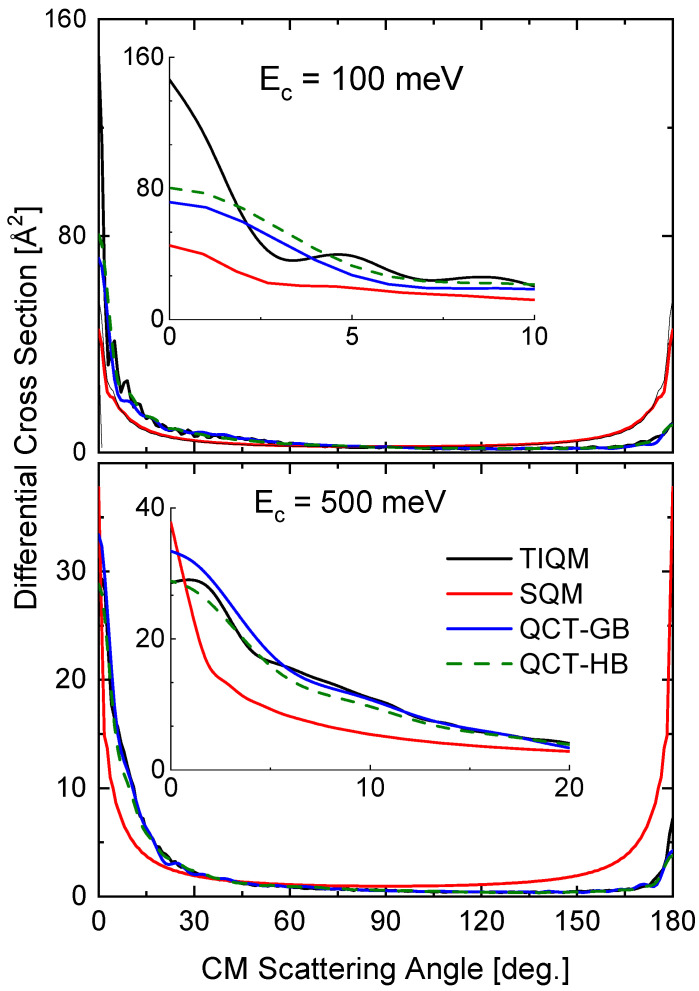
DCSs (in Å${}^{2}$) for the Ne + HeH${}^{+}\left(v,j\right)\to$ He + NeH${}^{+}\left(v^{\prime },j^{\prime }\right)$ reaction at two different values of the collision energy: 100 meV (**top panel**) and 500 meV (**bottom panel**). Comparison of TIQM (black lines), SQM (red lines) and QCT with GB (blue lines) and HB (dashed green lines) results from Ref. [64].

**Figure 14 molecules-26-04206-f014:**
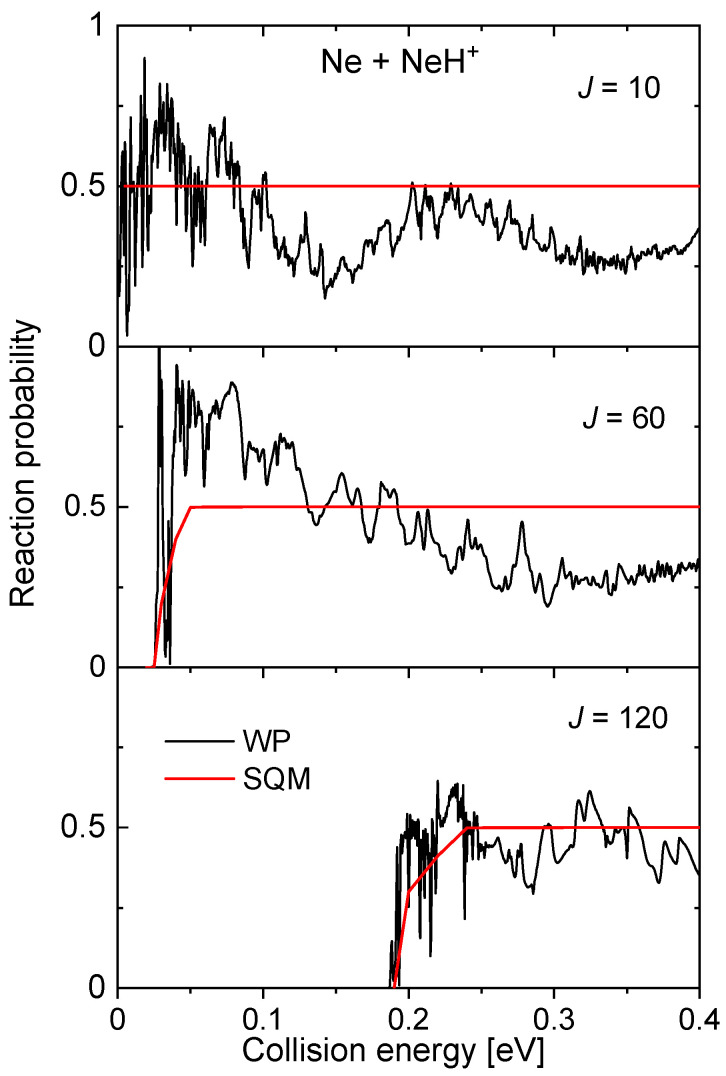
Reaction probability calculated as a function of the collision energy (in eV) for the Ne + NeH${}^{+}\to$ Ne + NeH${}^{+}$ reaction for three different values of the total angular momentum J=10 (**top panel**), J=60 (**middle panel**) and J=120 (red) obtained with the WP (red lines) and SQM (black lines) from Reference [70].

**Figure 15 molecules-26-04206-f015:**
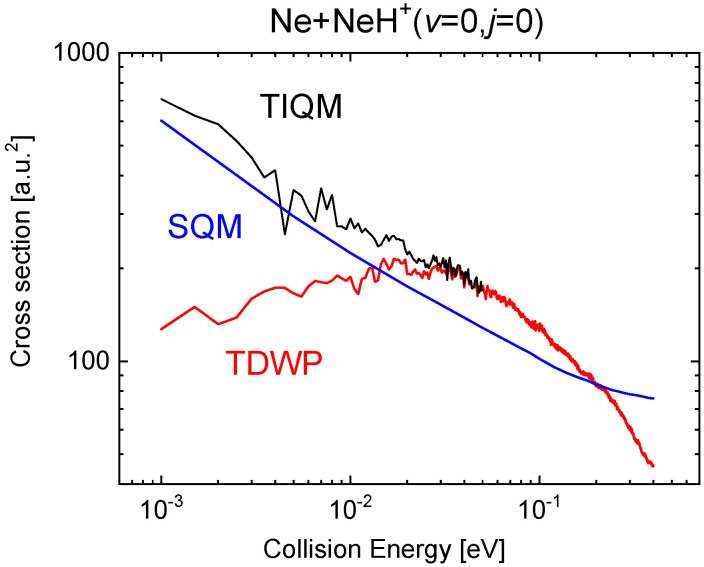
Cross sections as a function of the collision energy (in eV) for the Ne + NeH${}^{+}\to$ Ne + NeH${}^{+}$ calculated with the TIQM (black line), TDWP (red line), and SQM (blue line) methods in [70].

**Figure 16 molecules-26-04206-f016:**
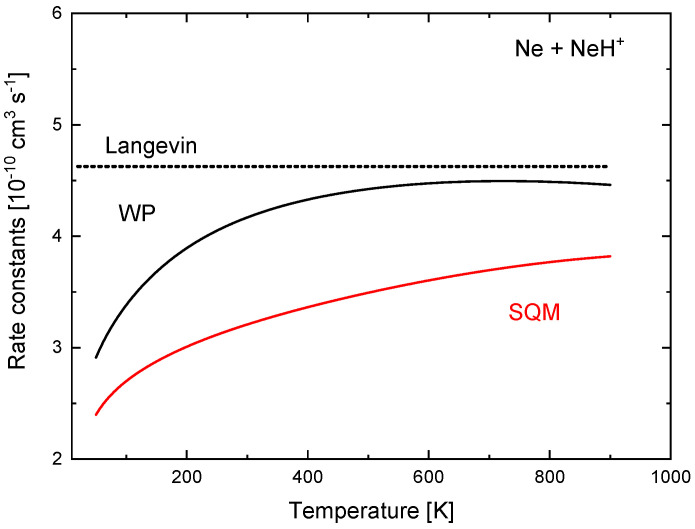
Rate constants (in 10${}^{-10}$ cm${}^{3}$ s${}^{-1}$) for the Ne + NeH${}^{+}$ reaction calculated with TDWP (black line) and SQM (red line) approaches of Reference [70]. Langevin prediction is also included (dotted line).

**Figure 17 molecules-26-04206-f017:**
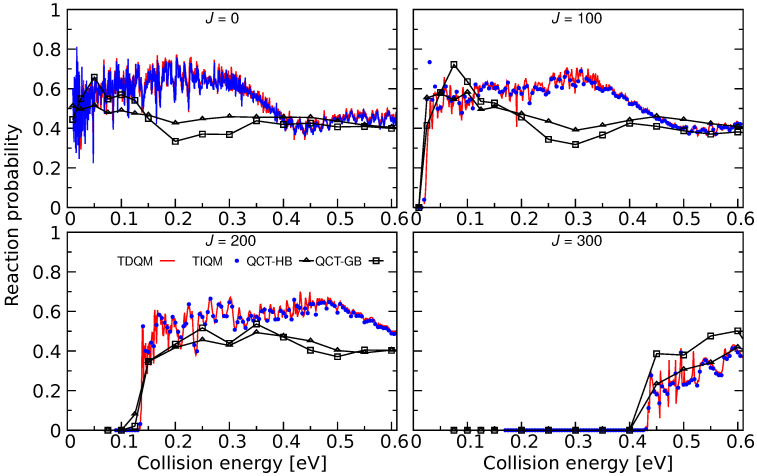
Total reaction probabilities as a function of collision energies for the Ar + ArH${}^{+}\left(v=0,j=0\right)$→ ArH${}^{+}$ + Ar for J=0, 100, 200, and 300 computed following TIQM (blue solid lines and symbols), TDQM (red solid lines), and QCT (QCT-HB: green lines+symbols, QCT-GB: black lines+symbols) methodologies. Data taken from Ref. [71].

**Figure 18 molecules-26-04206-f018:**
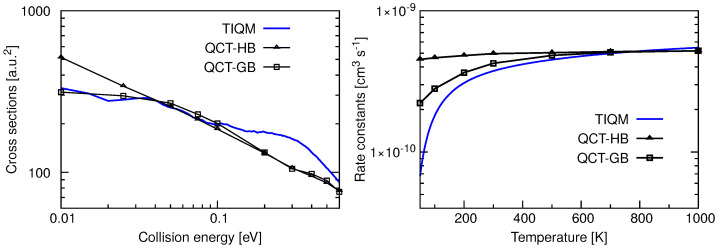
Total integral cross sections as a function of collision energy (**left panel**) and rate constants as a function of temperature (**right panel**) for the Ar + ArH${}^{+}\left(v=0,j=0\right)$→ ArH${}^{+}$ + Ar calculated following TIQM (solid blue line) and QCT (QCT-HB: green line+symbols, QCT-GB: black line+symbols) methods. Data are taken from Ref. [71].

**Table 1 molecules-26-04206-t001:** Details of methods and basis sets used in the electronic structure calculations, number of ab initio energies used in the fitting, and root mean square errors (RMSE) of the fitting in kcal/mol.

System	Ab Initio Method and Basis Set	Number of Ab Initio Energies	RMSE	Ref.
He${}_{2}$H${}^{+}$	CCSD(T)/d-aug-cc-pVTZ	2650	0.08	[34]
${\left[\mathrm{HeHNe}\right]}^{+}$	CCSD(T)/aug-cc-pVQZ	19,605	0.03	[69]
Ne${}_{2}$H${}^{+}$	CCSD(T)/aug-cc-pVQZ	22,537	0.026	[70]
Ar${}_{2}$H${}^{+}$	CCSD(T)/aug-cc-pVQZ	13,940	0.057	[71]

**Table 2 molecules-26-04206-t002:** Details of the global minima and reactant/product asymptotes obtained from the analytical PESs. The zero of energy is set to the energy of Rg+Rg’/Rg+H${}^{+}$ for each case.

${\left[\mathrm{HeHHe}\right]}^{+}$	$r_{\mathrm{eq}}$(HeH${}^{+}$) (a${}_{0}$)	1.746
	∠HeHHe	180.0°
	Energy (kcal/mol)	−60.319
He+HeH${}^{+}$	$r_{\mathrm{eq}}$(HeH${}^{+}$) (a${}_{0}$)	1.466
	Energy (kcal/mol)	−46.935
${\left[\mathrm{HeHNe}\right]}^{+}$	$r_{\mathrm{eq}}$(HeH${}^{+}$) (a${}_{0}$)	1.804
	$r_{\mathrm{eq}}$(NeH${}^{+}$) (a${}_{0}$)	2.102
	∠HeHNe	180.0°
	Energy (kcal/mol)	−65.447
Ne+HeH${}^{+}$	$r_{\mathrm{eq}}$(HeH${}^{+}$) (a${}_{0}$)	1.464
	Energy (kcal/mol)	−47.066
He+NeH${}^{+}$	$r_{\mathrm{eq}}$(NeH${}^{+}$) (a${}_{0}$)	1.872
	Energy (kcal/mol)	−53.433
${\left[\mathrm{NeHNe}\right]}^{+}$	$r_{\mathrm{eq}}$(NeH${}^{+}$) (a${}_{0}$)	2.156
	∠NeHNe	180.0°
	Energy (kcal/mol)	−70.038
Ne+NeH${}^{+}$	$r_{\mathrm{eq}}$(NeH${}^{+}$) (a${}_{0}$)	1.872
	Energy (kcal/mol)	−53.433
${\left[\mathrm{ArHAr}\right]}^{+}$	$r_{\mathrm{eq}}$(ArH${}^{+}$) (a${}_{0}$)	2.846
	∠ArHAr	180.0°
	Energy (kcal/mol)	−109.513
Ar+ArH${}^{+}$	$r_{\mathrm{eq}}$(ArH${}^{+}$) (a${}_{0}$)	2.423
	Energy (kcal/mol)	−93.963

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
