# Peer review of "Atom–Diatom Reactive Scattering Collisions in Protonated Rare Gas Systems"

_molecules, 2021, doi:10.3390/molecules26144206_

Round 1
Reviewer 1 Report
The review paper is written well and a detailed description of the methods and results are provided for an atom - diatom collisions in protonated rare gas systems.
I have just a few suggestions for the authors
It would be nice if the authors could include something (maybe a paragraph at the end) about the future directions and challenges in this area of study which I find somewhat missing in the paper.
Also, the Authors have shown many calculations for different reactions using various methods, but have not recommended which calculations may be good for further use in different models.
Overall I think once the authors include suggestions as mentioned, the paper can be accepted for publication.
Reviewer 2 Report
In this work authors present the study of the dynamics of atom-diatom reactions i.e. atom-diatom reactive scattering collisions. The study of those processes can provide useful information about the evolution of the stellar medium and generally is of astrophysical importance.
This contribution is a kind of review and I think that authors should change the type of the manuscript from Article to Review as they highlighted in the Abstract “In this work we review the state-of-art of the study of these…”.
The authors should also clearly state (in Abstract, Introduction and Conclusion) the main aim of this text. If nothing is new in the text the authors should beater emphasize why the text important.
Also, few technical shortcomings are in the text. Therefore, I cannot recommend this for publication at this time.
line 8: collisions => collisions.
According to the journal instruction for authors, Section Method should be placed after Section Results and Section Discussion.
line 154: "...a tridiagonal matrix as as ..."=>"...a tridiagonal matrix as ..."
line 234: "... equations is is perfomed..." => "...equations is performed ...."
line 348: "...and three posible..." => "...and three possible..."
caption of the Figures 12 and 13: "... collisione energy..." => "... collision energy..."
Caption of Figure 14: "... Reaction probabilty..." => "... Reaction probability..."
…etc. Authors should read the whole text carefully and correct minor flaws and typos.
Subsection 3.2 consists of only one subsubsection. 3.2.1. It should be corrected.
Conclusions need to be reworked and important things pointed out. It should be short precise and concise.
In Section Data Availability Statement, Acknowledgments and Conflicts of Interest: please remove text from template and fill with proper statements.
In References: If you use journals abbreviation than make it uniform. Frontiers in Chemistry => Front. Chem.
Ref [97]: [https://doi.org/10.1063/1.452944]. doi:10.1063/1.452944. => doi:10.1063/1.452944.
Ref. [101]: doi:{10.1021/jp902336s}. => doi: 10.1021/jp902336s.
There are a lot of references and it would be good to check them in detail.
